# Coupling between intra- and intermolecular motions in liquid water revealed by two-dimensional terahertz-infrared-visible spectroscopy

Maksim Grechko[1], Taisuke Hasegawa[1], Francesco D'Angelo[1], Hironobu Ito[2], Dmitry Turchinovich [1,3], Yuki Nagata[1] & Mischa Bonn[1]

The interaction between intramolecular and intermolecular degrees of freedom in liquid water underlies fundamental chemical and physical phenomena such as energy dissipation and proton transfer. Yet, it has been challenging to elucidate the coupling between these different types of modes. Here, we report on the direct observation and quantification of the coupling between intermolecular and intramolecular coordinates using two-dimensional, ultra-broad-band, terahertz-infrared-visible (2D TIRV) spectroscopy and molecular dynamics calculations. Our study reveals strong coupling of the O-H stretch vibration, independent of the degree of delocalization of this high-frequency mode, to low-frequency intermolecular motions over a wide frequency range from 50 to 250 cm$^{-1}$, corresponding to both the intermolecular hydrogen bond bending ($\approx 60$ cm$^{-1}$) and stretching ($\approx 180$ cm$^{-1}$) modes. Our results provide mechanistic insights into the coupling of the O-H stretch vibration to collective, delocalized intermolecular modes.

[1] Department of Molecular Spectroscopy, Max Planck Institute for Polymer Research, Ackermannweg 10, D-55128 Mainz, Germany. [2] Department of Chemistry, Faculty of Education, Shizuoka University, 836 Ohya, 422-8529 Shizuoka, Japan. [3] Fakultät für Physik, Universität Duisburg-Essen, Lotharstr. 1, 47057 Duisburg, Germany. Correspondence and requests for materials should be addressed to M.G. (email: grechko@mpip-mainz.mpg.de)

Water is the most common solvent in nature and industry, accommodating many chemical reactions, with water molecules providing a unique environment for reactivity of dissolved (bio-)molecules. Aqueous solutions are highly dynamic: the network of strong hydrogen bonds (HB) between the water molecules rearranges on (sub-)picosecond time scales[1–8]. These dynamics are driven by different types of thermally excited intermolecular motions, which make up the low-frequency modes (LFM) of liquid water. Recently, a growing number of studies point to strong mixing between the LFM and high-frequency intramolecular modes (HFM) of water, implying an active role of such mixed states in the chemistry of water[9–11]. This emerging picture of water dynamics makes the quantitative characterization of the mixing between the LFM and HFM states crucial for a fundamental understanding of this ubiquitous liquid. To this end, we develop a two-dimensional terahertz-infrared-visible (2D TIRV) spectroscopy, which can directly measure the coupling between the LFM and HFM vibrations. We use the 2D TIRV spectroscopy to measure the coupling between the LFM and O-H stretch vibrations in liquid water.

Quantification of coupling between different vibrational modes requires a two-dimensional (2D) spectroscopy that can measure the cross-peaks between the corresponding vibrations. So far, several types of the 2D vibrational spectroscopy have been introduced: 2D-Infrared (IR)[12–17], 2D-DOVE[18], 2D-Raman[19,20], 2D-THz-THz[21–23], 2D-Raman-THz[24–26] and 2D-THz-THz-Raman[27]. These approaches excite and probe couplings of the vibrational modes in a very similar frequency ranges; 2D-IR and 2D-DOVE probe the vibrational couplings between high-frequency modes such as O-H stretch modes, while the other four techniques probe the coupling amongst LFMs. Since the high-frequency modes and LFM vibrate at infrared and terahertz frequencies, respectively, elucidating the coupling between these two would require the implementation of 2D spectroscopy with infrared and terahertz pulses.

Here, we combine ultra-broadband THz pulses with the IR pulses and develop a 2D spectroscopy, which enables us to measure the direct coupling of the HFM ($>1000$ cm$^{-1}$) and LFM ($<500$ cm$^{-1}$). We explore the coupling between the water LFM with energies from 20 to 450 cm$^{-1}$ (Fig. 1a) and the O-H stretch mode with a vibrational frequency of $\sim 3400$ cm$^{-1}$ (Fig. 1b) and answer whether and how the O-H stretch mode is coupled to the LFM in liquid water. The O-H stretch mode is known to form a delocalized vibrational state in liquid H$_2$O[28–30]. That is, O-H stretch vibrations in individual molecules are coupled amongst each other and the vibrational eigenstate is given by a coherent superposition of individual molecular vibrations. Since dilution of H$_2$O in D$_2$O leaves the HB network and LFM intact, but decouples the O-H stretch oscillators[31], varying the concentration of O-H groups allows us to investigate how the coupling between the O-H stretch and LFM changes by the formation of vibrational excitons. By comparing the experimental 2D TIRV data with simulated 2D TIRV spectra, we clarify the nature of the vibrational coupling of O-H stretch mode and LFM.

## Results

**2D TIRV spectroscopy.** To quantify the coupling between the LFM and O-H stretch mode, we measure a signal enhanced by resonances with both of these vibrations. To this end, we develop nonlinear, 2D TIRV spectroscopy. This type of spectroscopy was recently proposed by Ito and Tanimura[32] and can be considered as an extension of the 2D-DOVE (IIV-SFG)[18,33] spectroscopy to the THz-IR spectral range. Figure 2a shows the experimental scheme of the 2D TIRV spectroscopy. We use three optical pulses at terahertz (THz), infrared (IR), and visible (VIS) frequencies. A

broadband ($\sim 250$ cm$^{-1}$ FWHM, Fig. 2b; see also Supplementary Note 1) THz pulse laser-generated in air plasma[34–36] creates a vibrational coherence state in a sample by excitation of its LFM (Fig. 2d). This vibrational coherence oscillates at THz frequencies and after a time delay $t$ interacts with simultaneously arriving broadband ($\sim 350$ cm$^{-1}$ FWHM, Fig. 2c) IR and narrowband ($\sim 30$ cm$^{-1}$ FWHM, centered at 12,500 cm$^{-1}$) visible laser pulses. The three (THz, IR, and VIS) interactions can cause the sample to emit new light at $\omega_{VIS} + \omega_{IR} \pm \omega_{THz}$ frequencies, as exemplified by the energy level diagram in Fig. 2d. This process makes up the well-known four-wave mixing (FWM). We use a spectrometer and camera to measure the FWM light emitted by the sample and employ heterodyne detection, allowing us to measure the electric field of the signal rather than its intensity. Heterodyne detection is essential for obtaining the THz frequency axis for the 2D spectra as explained below. To implement heterodyne detection, we add an additional laser pulse, a local oscillator (LO), which interferes at the detector with the signal wave from the sample. To derive the IR frequency axis, $\omega_2$, for the 2D TIRV spectra, we subtract the frequency of the VIS pulse from the frequency of the signal wave. To derive the THz frequency axis, $\omega_1$, we perform measurements in time-domain. That is, we measure signal as function of the time delay $t$ between the THz pulse and the IR/VIS pulse pair (Fig. 2a). The phase of the emitted signal field changes linearly with $t$ and is proportional to the frequency of the LFM coherence induced by the THz pulse. Fourier transform of the measured time-domain data generates the THz frequency axis for the 2D TIRV spectrum. In the figures we plot the $\omega_2$ and $\omega_1$ axes vertically and horizontally, respectively.

The interaction with the THz, IR, and VIS laser fields can excite different degrees of freedom in the material. The generation of the FWM signal is enhanced when states $|1\rangle$, $|2\rangle$, and $|3\rangle$ (Fig. 2d) are coupled, similar to the 2D-DOVE spectroscopy[18,33,37]. The coupling between the states can stem from the mechanical and/or electrical anharmonicity[32,33]. We note that in general, the off-diagonal peaks in a doubly vibrationally

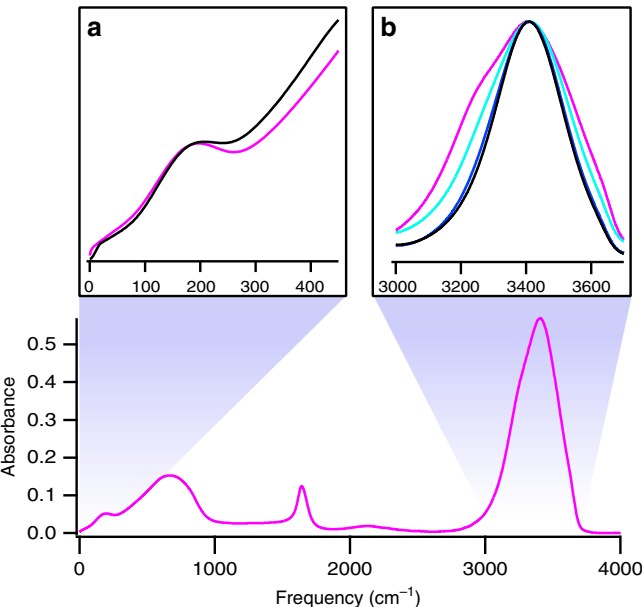

**Fig. 1** The infrared response of bulk liquid water. The absorption spectrum for H$_2$O at 25 °C in the 0–4000 cm$^{-1}$ frequency range (bottom panel). **a** absorbance for H$_2$O (magenta) and D$_2$O (black) in the far-infrared spectral range. **b** The normalized absorbance for H$_2$O (magenta), 50% H/D (turquoise), 20% H/D (blue), and 5% H/D (black) in the mid-infrared spectral range. In bottom panel and (**a**) we use data from ref. [47]

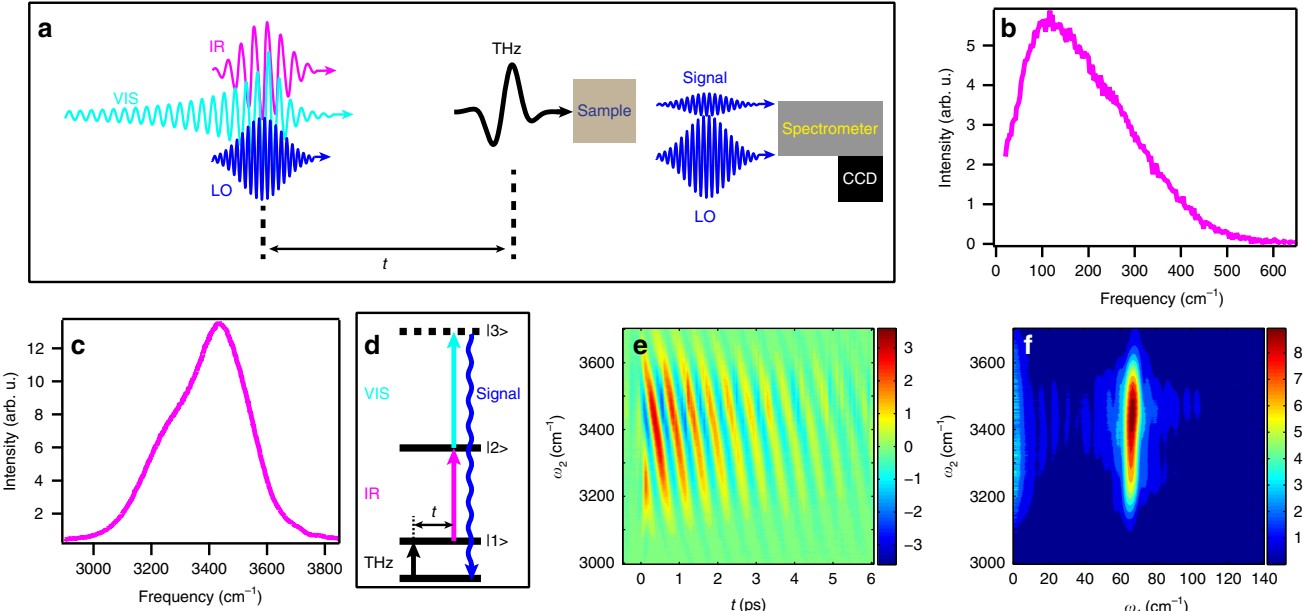

**Fig. 2** Details of the experimental approach. **a** Schematics of the 2D TIRV experiment. FWM of the terahertz (THz, black), infrared (IR, magenta), and visible (VIS, turquoise) laser pulses in a sample produces signal wave (blue), which is spectrally resolved and interferes with the local oscillator (LO, also blue) at the detector (CCD camera). The interference is measured for different time delays $t$ between the IR/VIS pair and the THz pulse. **b**, **c** Intensity spectra of the THz and IR laser pulses, respectively. **d** Energy level diagram for the FWM employed for the 2D TIRV spectroscopy. **e** Time-domain 2D TIRV data for $CaF_2$. **f** 2D TIRV spectrum for $CaF_2$ produced by Fourier transform of the time-domain data in (**e**)

enhanced spectroscopy can also be generated by excitation of the overtone and population states of the same mode[32,38], but in 2D TIRV spectroscopy such excitation pathways are impossible due to the large frequency difference between the THz and IR fields. Thus, the off-diagonal peaks in the 2D TIRV spectra reflect coupling between different modes. The principle of the 2D TIRV spectroscopy is demonstrated using the model sample $CaF_2$. For $CaF_2$ the THz pulse resonantly excites phonon modes and oscillation of the excited phonon is measured by the IR and VIS beams by promoting non-resonant transitions to virtual states. The time-domain data contain a signal that oscillates with the period of ~500 fs and decays on a time scale of few picoseconds (Fig. 2e). The corresponding peak in the 2D TIRV spectrum can be found at $\omega_1 = 67 \, cm^{-1}$ (Fig. 2f). This resonance can be assigned to the transition between (thermally excited) $T_{1u}$ and $T_{2g}$ phonon bands of the $CaF_2$[39]. The 2D TIRV peak is sharp along the THz axis and broad along the IR axis, which agrees with the notion of resonant and non-resonant excitations, respectively. The linewidth of the peak along the abscissa is determined by the lifetime of the phonon coherence, while its ordinate linewidth is limited by the bandwidth of the IR laser pulse.

**Experimental 2D TIRV spectroscopy results for water**. To elucidate the coupling between the intramolecular and intermolecular degrees of freedom in liquid water, we measure 2D TIRV spectra for $D_2O$, $H_2O$ and 5, 20, and 50% (volume fraction) of $H_2O$ in $D_2O$. The time-domain 2D TIRV results are shown in Fig. 3a–e. For $D_2O$, the signal is generated by resonant interaction with the THz pulse (Fig. 1a, Supplementary Fig. 1 shows the Liouville excitation pathways). Because $D_2O$ exhibits no transitions in the O-H stretch frequency range of our IR laser pulse, we term this signal singly resonant. The coherence induced by the THz pulse in $D_2O$ has an intricate shape and has fully decayed by ≈130 fs (Fig. 3a, see Methods for the definition of $t = 0$). For 5% H/D blend consisting of $D_2O$ (90.25%), HOD (9.5%), and $H_2O$ (0.25%) molecules, the O-H stretch oscillators of HOD molecules

are largely decoupled from each other due to the high degree of dilution[31]. Therefore, O-H stretch vibrations are largely isolated and localized in individual HOD molecules. The O-H stretch vibration in HOD is resonant with the IR laser field and generates the signal through resonant interactions of both THz and IR pulses. This new doubly resonant response is apparent in the time-domain data at $\omega_2 \approx 3450 \, cm^{-1}$ and decays more slowly, with signal appearing up to $t \approx 240$ fs (Fig. 3b). Upon increase of the $H_2O$ concentration to 20%, the doubly resonant signal increases significantly and obscures the singly resonant O-D signal (Fig. 3c). Further increase of the $H_2O$ concentration broadens the response along the IR axis and at the same time the signal intensity decreases (Fig. 3d, e). The broadening of the signal to lower IR frequencies is consistent with the change of the linear absorption spectrum (Fig. 1b). The decay time of the coherent motion of the LFM coupled to O-H stretch is independent on the H/D ratio and remains $t \approx 240$ fs.

For the water signals depicted in Fig. 3a–e to be generated by FWM of THz, IR, and VIS pulses it must be produced by one interaction of the sample with each of these fields. Because we detect the signal at the sum frequency of the IR and VIS laser pulses ($\omega_{VIS} + \omega_{IR} \pm \omega_{THz}$, ~15,800 $cm^{-1}$) the signal must indeed be generated by one interaction with each of the IR and VIS fields. The linear dependence of the signal on the strength of the THz field (Supplementary Fig. 2) confirms that a sample interacts only once with the THz pulse, in line with these 2D TIRV signals being generated by FWM of the THz, IR and VIS fields in water.

Because for water the decay time of the LFM coherence is comparable to the duration of the THz pulse, the time-domain 2D TIRV data are less intuitive than the frequency-domain 2D TIRV spectra. A 2D TIRV spectrum represents the product of the spectrum of the third-order nonlinear response function and the spectra of the laser pulses (Supplementary Note 2). Absolute-value 2D TIRV spectra produced by Fourier transform of the time-domain data are shown in Fig. 3f–j. For the pure $D_2O$ and the 5% H/D mixture, substantial signal intensity appears along the $\omega_1$ axis with frequencies up to 350 $cm^{-1}$. This is in a marked

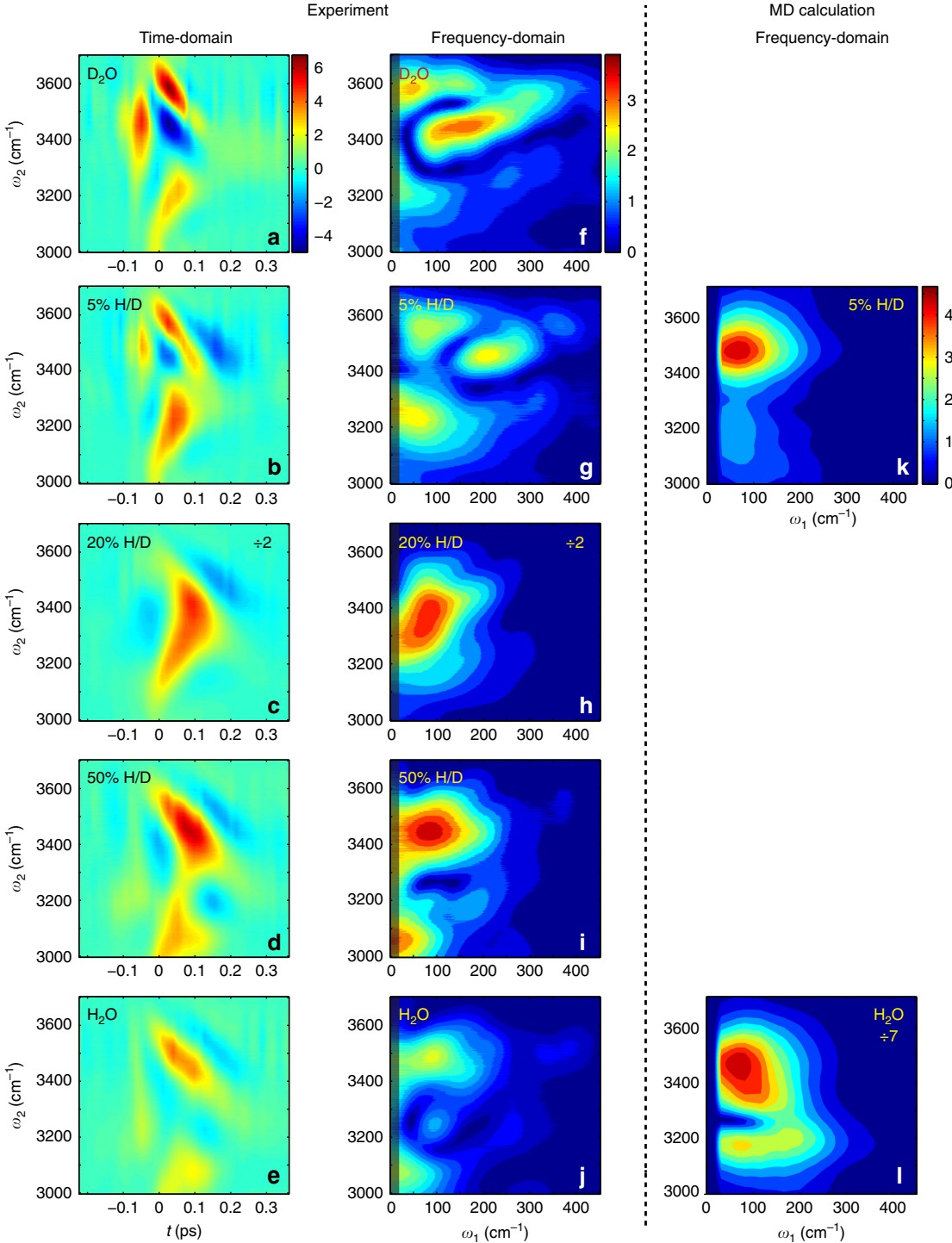

**Fig. 3** Time-domain and frequency-domain 2D TIRV data for differently isotopically diluted water. The plots in the left column show time-domain data for 100% $D_2O$ (**a**), 5% H/D (**b**), 20% H/D (**c**), 50% H/D (**d**), and 100% $H_2O$ (**e**) samples. The plots in the middle column (**f–j**) show the absolute-value 2D TIRV spectra obtained by Fourier transform of the time-domain data in (**a–e**), respectively. Because of the weak THz intensity below 20 cm$^{-1}$ and high sensitivity of this spectral range to the baseline fluctuations in the time-domain data the 2D TIRV spectra for $\omega_1 < 20$ cm$^{-1}$ are not reliable. Thus, we shade this spectral range in plots (**f–j**). The plots in the right column (**k, l**) show the absolute-value 2D TIRV spectra calculated using MD simulations and convoluted with the THz and IR laser pulses

contrast to the other samples. For the 20% H/D mixture the highest-frequency LFM contributing noticeably to the spectrum have frequencies of ≈200 cm$^{-1}$. This observation confirms that for 20% H/D the singly resonant contribution from the O-D groups to the signal is negligible. Thus, the spectra in Fig. 3h–j are generated via double resonance by the O-H groups.

**Molecular dynamics calculations**. Figure 3k, l show the absolute-value 2D TIRV spectra obtained by classical molecular dynamics (MD) calculations for the 5% H/D and 100% $H_2O$ samples, respectively. The calculations of the nonlinear optical susceptibility are performed analogously to those in ref. [32]. In brief, we use POLI2VS water model[40] for 64 $H_2O$ and 58 $D_2O$+6 HOD

molecules representing the 100% $H_2O$ and 5% H/D samples, respectively. After 100 ps equilibration MD runs under the constant temperature conditions at 300 K, the time-domain 2D TIRV response functions are calculated for the time periods of 250 fs using the non-equilibrium–equilibrium hybrid response function algorithm using a total of $10^6$ non-equilibrium trajectories. The 2D TIRV spectra are generated by Fourier transforming the time-domain response functions and subsequent convolution in the frequency domain with the THz and IR pulses of the experiment to allow direct comparison with the experimental results (see Supplementary Methods for more details on calculations and the spectra without the convolution). Because the calculations are time-demanding, we report here the spectra for only two samples which represent the limiting cases of very low (5%) and very high (100%) O-H concentration. The 2D TIRV spectrum of the 5% H/D mixture consists of an O-H signal (most prominent at $\omega_2 \sim 3500$ $cm^{-1}$), interfering with an O-D response (most prominent at $\omega_2 \sim 3150$ $cm^{-1}$). The latter signal is the weak tail of the O-D stretch resonance centered at $\sim 2500$ $cm^{-1}$. The calculated 100% $H_2O$ spectrum has a prominent nodal line at 3250 $cm^{-1}$, which is consistent with the minimum of the signal in the measured 100% $H_2O$ spectrum. The qualitative agreement between the experimental and simulation data in Fig. 3 is reasonable, yet there are reproducible features in the data that are not captured by the simulations (discussed in more detail below).

## Discussion

Both experimental and simulation data clearly indicate substantial coupling between the high-frequency O-H stretch and the LFM of liquid water. In the experiment, the doubly resonant 2D TIRV spectrum of O-H strongly depends on the concentration of the O-H groups. For the 5% H/D mixture the singly resonant O-D signal is approximately equal to the doubly resonant O-H signal, with the two signal contributions interfering at $\omega_2 = 3400$ $cm^{-1}$ (Fig. 3a, b, f, g). The extraction of the measured O-H signal for this sample is discussed below. For the 20% H/D mixture, the signal intensity is very strong, and the spectrum is primarily composed of a single peak centered at $\omega_1 = 80$ $cm^{-1}$, $\omega_2 = 3370$ $cm^{-1}$ (Fig. 3c, h). Further increasing the H/D ratio to 50% results in a decrease of the signal intensity and a marked change of the lineshape of the 2D TIRV spectrum (Fig. 3d, i). The signal intensity for different sample compositions depends on the density of resonant O-H groups, coupling strength between O-H stretch and LFM, attenuation of the IR and THz beams, wavevector mismatch, and possible interference of different contributing signals. These effects are discussed in more detail in the Supplementary Note 3.

The most prominent trait of the new lineshape observed for H/D ratio of 50% is a nodal line (signal minimum) around $\omega_2 = 3270$ $cm^{-1}$. This change must originate from the modification of the molecular third-order optical response function $\langle S_{OH}^{(3)} \rangle$, responsible for the FWM process. Because the change of the LFM with isotope dilution is minute (Fig. 1a), the drastic change of the $\langle S_{OH}^{(3)} \rangle$ must be attributed to a change in the character of the O-H stretch vibrations. Increasing concentration of the O-H groups begets two new types of vibrational states. Firstly, within each $H_2O$ molecule, the O-H stretch is mixed with the overtone vibration of the H-O-H bending mode because of the Fermi resonance (FR). This resonance is appreciably weaker for HOD because of the large energy mismatch of the two levels[41]. Thus, increasing number of $H_2O$ species enhances signal generated by the FR. Secondly, the increasing density of the O-H groups induces the formation of the delocalized O-H stretch exciton because of intermolecular coupling. As a result, the change of the

$\langle S_{OH}^{(3)} \rangle$ can be linked to the appearance of the FR and vibrational exciton formation.

For the 100% $H_2O$ sample, the 2D TIRV signal intensity is reduced as compared to the 50% H/D mixture (Fig. 3e, j). The decrease of the signal is accompanied by a further evolution of the lineshape in the 2D TIRV spectrum. This change is characterized mainly by the appearance of a peak at $\omega_1 = 95$ $cm^{-1}$, $\omega_2 = 3250$ $cm^{-1}$, which was the position of the nodal line for the 50% H/D sample.

For the 5% H/D sample, the signal intensities originating from the O-H and O-D groups are comparable (Fig. 3f, g). To extract the 'pure' O-H response from the 5% H/D signal, we seek to eliminate the O-D contribution. To this end, we use the 2D TIRV spectrum for 100% $D_2O$ and consider the dependence of the signal intensity on the sample composition. This dependence stems from the different linear absorption coefficients and wavevector mismatch of the laser beams in the samples. The appropriately weighted difference of the two (complex-value) spectra (see Supplementary Note 3 for detailed calculation) is displayed in Fig. 4a. We attribute this 2D TIRV spectrum to that of the isolated O-H stretch oscillator, $S_{loc}$. $S_{loc}$ is dominated by a peak centered at $\omega_1 = 100$ $cm^{-1}$, $\omega_2 = 3415$ $cm^{-1}$. The spectrum $S_{loc}$ derived from the 5% H/D response closely resembles the 2D TIRV spectrum of the 20% H/D sample (Fig. 4b, see also Supplementary Fig. 3 for comparison of the time-domain data). The similarity between the corrected 5% spectrum and the 20% H/D spectrum suggests that the 20% H/D spectrum ($D_2O$:HOD:$H_2O$ ratio of 16:8:1) is still dominated by isolated O-H stretch vibrations and the contribution of the coupled oscillators is small. Because of the better signal-to-noise ratio, we from here on assume $S_{loc}$ to be equal to the 20% spectrum.

After characterizing the coupling between the LFM and the localized O-H stretch oscillators we aim to determine how this coupling changes with the excitonic delocalization of the latter. Based on the similarity between the 50% H/D and 100% $H_2O$ spectra, we hypothesize that the spectrum for the 50% H/D blend can be decomposed into contributions from $S_{loc}$ and from coupled O-H stretch oscillators. Indeed, subtracting $0.23 \times S_{loc}$ from the 50% spectrum reproduces the spectrum measured for 100%

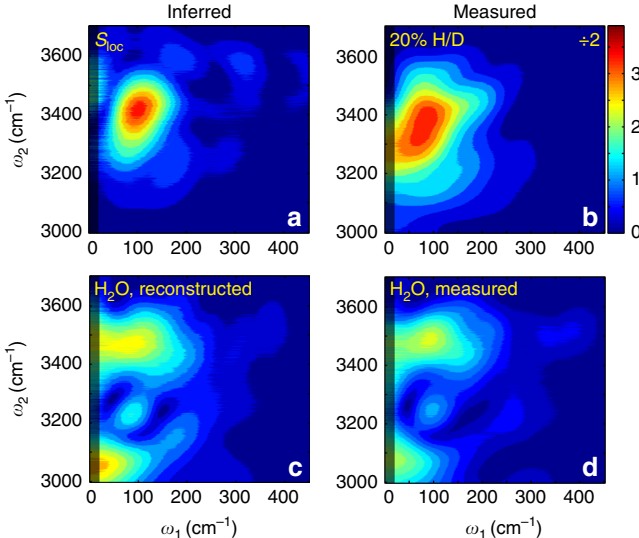

**Fig. 4** Inferred response for isolated and coupled O-H oscillators. Absolute-value 2D TIRV spectra (**a**), $S_{loc}$ for isolated O-H oscillators derived from the 5% spectrum in Fig. 3g (**c**), reconstructed for 100% $H_2O$ by subtracting the 50 and 20% H/D spectra. For comparison, the 20 and 100% spectra from Fig. 3 are reproduced in panels (**b**) and (**d**)

$H_2O$ (Fig. 4c, d and Supplementary Fig. 3), where the amplitude 0.23 was simply chosen to optimize the agreement. This finding implies that for all isotope dilutions the 2D TIRV spectra can be described, within the signal-to-noise of our measurements, as a linear combination of two signals. That the spectra can be described by two components needs not imply that there are strictly two types of O-H groups (those with local vibrations, and those with FR/exciton vibrations). Still, the invariance of the FR/exciton lineshape for 50% H/D and 100% $H_2O$ samples indicates that the spectrum of the LFM coupled to the O-H stretch vibration is invariant to the delocalization of the latter.

To further investigate the nature of the LFM that the high-frequency O-H stretch couples to, we compare the experimental results with the results obtained by the MD simulations. We first note that a comparison of the 2D TIRV spectra in Fig. 3 reveals that the experimental spectra contain more structure than the calculated spectra. Particularly, for the 5% H/D mixture the measured O-D spectrum around $\omega_2 = 3500\ cm^{-1}$ is more structured and extends to the higher $\omega_1$ frequencies. Also, while the calculated spectrum for the 100% $H_2O$ has only a nodal line at $\omega_2 = 3250\ cm^{-1}$, which is similar to the measured 50% H/D spectrum, the measured 100% $H_2O$ spectrum has an additional small peak at this frequency. The origin of these dissimilarities is not clear yet and requires further investigation. Apart from these dissimilarities the experimental and calculated data are in good qualitative agreement, which allows us to gain mechanistic insight into the coupling between the O-H stretch and the vibrations of the LFM.

Analysis of the 2D TIRV spectra in terms of the frequencies of the coupled oscillators is complicated by interfusion of the real and imaginary parts of their responses to the optical excitations. Although the real part of the oscillator response to the optical excitation is related to the refraction of light and has a dispersive lineshape, the imaginary part of the response is related to the absorption of light and has an absorptive lineshape. Thus, an absorptive-like spectrum of the third-order optical response measured by the 2D TIRV spectroscopy gives a more intuitive representation of the frequencies of the coupled oscillators.

The absorptive-like 2D TIRV spectra can be readily produced for the calculated data by performing the *sin–sin* Fourier transform of the time-domain response (see Supplementary Methods for details)[24]. To obtain a genuine representation of the coupled high-frequency and low-frequency oscillators, we do not convolve the *sin–sin* 2D TIRV spectra with the laser pulses. Figure 5a, b show the *sin–sin* 2D TIRV spectra calculated for the 5% H/D and 100% $H_2O$, respectively. The spectrum for the 5% H/D is dominated by a peak centered at $\omega_1 = 135\ cm^{-1}$, $\omega_2 = 3395\ cm^{-1}$. This peak has 230 $cm^{-1}$ linewidth along the $\omega_1$ frequency axis and can be readily assigned to the O-H stretch coupled to both the HB bending (at ≈60 $cm^{-1}$) and stretch (at ≈180 $cm^{-1}$) motions[42–44]. The *sin–sin* Fourier transform eliminates the singly resonant O-D response in the 5% H/D spectrum. The *sin–sin* 2D TIRV spectrum of the 100% $H_2O$ is characterized by a peak virtually identical to 5% H/D along the $\omega_1$ axis, but it is centered at $\omega_2 = 3250\ cm^{-1}$. This red shift of the maximum in the *sin–sin* 2D TIRV spectrum of 100% $H_2O$ along the $\omega_2$ axis is in a stark contrast with the linear IR spectra (Fig. 5c). For both 5 and 100% $H_2O$ the maximum of the $H_2O$ linear absorption is located at ≈3400 $cm^{-1}$, i.e., the O-H stretch peak frequency is insensitive to the concentration of $H_2O$. Thus, the maximum in the *sin–sin* 2D TIRV spectrum at $\omega_2 = 3250\ cm^{-1}$ indicates that in pure $H_2O$ the O-H stretch modes at ~3250 $cm^{-1}$ have stronger coupling to the intermolecular vibrations as compared to the modes at ~3400 $cm^{-1}$.

Based on the agreement between the experimental and theoretical data, we draw the following conclusions about the coupling between the O-H stretch and LFM in liquid water:

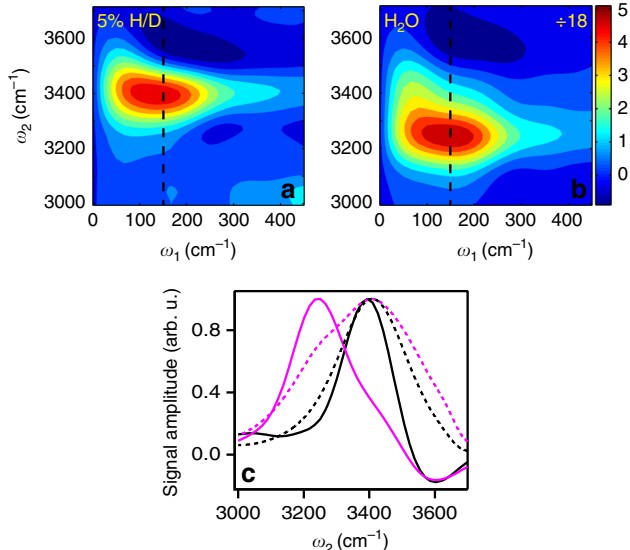

**Fig. 5** Absorptive-like 2D TIRV spectra for differently isotopically diluted water. The spectra for 5% H/D (**a**) and 100% $H_2O$ (**b**) are obtained by the *sin–sin* Fourier transform of the time-domain data from the MD simulations. The solid black and magenta lines in plot (**c**) show slices of the spectra in (**a**) and (**b**), respectively. The slices are taken at $\omega_1 = 150\ cm^{-1}$ (vertical dashed lines in (**a**) and (**b**)). The dashed black and magenta lines in (**c**) show absorbance of 5% H/D and 100% $H_2O$ samples, respectively. All spectra in (**c**) are normalized at their maxima

I.  The qualitative agreement between the calculated and experimental 2D TIRV spectra for the 100% $H_2O$ indicates that the O-H stretch is directly coupled to the 50–250 $cm^{-1}$ LFM. Based on the assignment of the far-IR linear absorption spectrum of liquid water, this means that the O-H stretch is coupled to both the HB bending (≈60 $cm^{-1}$) and HB stretch (≈180 $cm^{-1}$) intermolecular modes.

II. Both the experimental and calculated data evidence that the spectrum of the LFM coupled to the O-H stretch vibration is invariant to the O-H stretch delocalization.

III. In both the experimental and calculated 2D TIRV spectra, we observe a shift of the sample response to the lower IR frequencies with an increase of the $H_2O$ concentration. Thus, for the 100% $H_2O$ system, the coupling is markedly stronger for the lower frequency, ~3250 $cm^{-1}$, O-H stretch modes.

The mechanism of the vibrational relaxation in liquid water has been the subject of intensive debate, with the usual focus on the sequential conversion of the O-H stretch excitation to heat via the H-O-H bending and librational modes[45]. Our study unambiguously reveals direct, strong coupling of both the localized and delocalized O-H stretch vibration to the hydrogen bond stretch and bending modes, despite the very large frequency mismatch between the high-frequency and low-frequency modes. Our findings are in good agreement with the recent theoretical predictions by Ito and Tanimura using the same MD model[32]. Their analysis shows that for water both electrical and mechanical anharmonicities contribute significantly to the coupling. Strong mechanical coupling between the O-H stretch and the LFM manifests that in the liquid the low-frequency intermolecular and high-frequency intramolecular motions generate cooperative motion of the atoms. Such cooperative motion provides a competing channel for direct non-adiabatic energy dissipation to collective, delocalized intermolecular modes. As such, our study strongly suggests that vibrational energy relaxation from the O-H

stretch mode to the LFM heat can occur through the direct coupling of these modes, in addition to the previously suggested stretch-bend exciton coupled to the LFM[10] and/or vibrational conical intersections[9,46].

In conclusion, the current work presents a spectroscopy technique to investigate the coupling between low-frequency intermolecular vibrational modes and high-frequency local modes. We applied it here to elucidate mode coupling in pure water, yet the technique can be readily extended to uncover the heterogeneity/ homogeneity of water near ions, osmolytes, and biomolecules, such as proteins.

## Methods

**Experimental setup**. Experimental setup for the 2D TIRV spectroscopy is shown in Supplementary Fig. 4. The output from a regenerative amplifier with a repetition rate of 1 kHz, a central wavelength of 800 nm and 60 nm FWHM (Spitfire Ace, Spectra-Physics) is split into three beams. The first beam with about 1 mJ/pulse energy pumps a traveling-wave optical parametric amplifier with non-collinear difference frequency generator (TOPAS prime, Light Conversion) to generate the IR laser pulse. The second beam with about 0.4 mJ/pulse energy is used to produce the VIS laser pulse. To this end its bandwidth is narrowed to 30 cm$^{-1}$ by passing the beam through an air spaced Fabry Perot etalon (FSR 1035 cm$^{-1}$, Fe ~ 35 @ 800 nm, SLS Optics Ltd). The third beam with about 1 mJ/pulse energy generates broadband THz pulse via two-color femtosecond laser mixing in air plasma[34]. We polarize the IR and VIS pulses horizontally and align them collinear by using a dichroic beam combiner (BC in Supplementary Fig. 4) (Laseroptik GmbH). The LO pulse is produced by IR and VIS beams at protected gold mirrors (M1 and M2 in Supplementary Fig. 4) by sum-frequency generation. The IR, VIS, and LO beams are focused by a CaF$_2$ lens (L1 in Supplementary Fig. 4) (f = 15 cm) through a hole in an off-axis parabolic mirror (PM in Supplementary Fig. 4). The focal plane for the IR beam is at the sample, whereas for the VIS beam it is ≈1.5 cm before the sample. The energies for the IR and VIS pulses at the sample are 0.3 µJ and 40 µJ, respectively. The THz pulse is polarized horizontally by a high contrast grid array polarizer (P1 in Supplementary Fig. 4) (P01, InfraSpecs) and focused onto the sample by parabolic mirror (PM in Supplementary Fig. 4). After the sample, the signal and LO beams are collimated by a CaF$_2$ lens (L2 in Supplementary Fig. 4) and aligned to the spectrometer (Acton SP 2300, Princeton Instruments). We detect horizontally polarized signal by employing a nanoparticle linear film polarizer (P2 in Supplementary Fig. 4) (LPVIS050, Thorlabs Inc). The signal is measured by EMCCD camera (Newton 970, Andor Technology Ltd). To vary the time delay t for the THz pulse, we use a motorized translation stage (TS in Supplementary Fig. 4) (M-521.DD, Physik Instrumente GmbH).

The area of the setup around the THz beam pathway and the sample was enclosed in a box and purged with dry nitrogen. The sample cell for water is composed of two windows separated by a 1 mm-thick Viton O-ring. The front window of the cell is stainless steel with a pinhole of 1 mm diameter. The back window is 2-mm-thick CaF$_2$ window. Water leakage from the sample cell through the pinhole in the front window is prevented by the surface tension and the laser beams are aligned through the pinhole. The IR and THz laser pulses are absorbed by the 1 mm water sample and do not reach the back window, which prevents CaF$_2$ signal in the 2D TIRV spectra of water.

2D TIRV spectra for CaF$_2$ are measured using a standard optical 2-mm-thick CaF$_2$ window. Time-domain data for CaF$_2$ display a short-lived non-resonant signal at frequencies $\omega_2 \le 2800$ cm$^{-1}$ (Supplementary Fig. 5), which is presumably generated by interaction of the THz pulse with electrons of the material. We use this non-resonant signal to determine time delay $t = 0$.

**Sample preparation**. The samples were prepared by mixing the volume fractions of Milli-Q H$_2$O (resistance 18.2 MΩ cm) and D$_2$O (99.90% D, Euriso-Top SAS) shortly before the measurements and were stored in a box purged with dry nitrogen. All measurements were performed at room temperature (23 °C) using horizontally polarized THz, IR, and VIS pulses and detecting horizontally polarized signal light.

**Data availability**. The authors declare that the data supporting the findings of this study are available within the paper and its Supplementary Information files or from the corresponding author on reasonable request.

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

## Acknowledgements

This work has been financially supported by the Max Planck Society.

## Author contributions

M.G. and M.B. conceived the study; M.G. and F.D'A. constructed the experimental setup; M.B. and D.T. provided equipment for the experimental setup; M.G. performed the experiments; T.H. and H.I. performed the MD simulations; M.G., T.H., Y.N. and M.B. analyzed the data and wrote the manuscript; and D.T. and F.D'A. commented on the manuscript.

## Additional information

**Competing interests:** The authors declare no competing interests.

