## [Peer Review File · Nature Communications]

Reviewer #1 (Remarks to the Author):

This paper reports an interesting and potentially powerful new spectroscopy for studying the dynamics of water by revealing the coupling of high frequency OH stretch vibrations with low frequency modes of water. The method, termed terahertz-infrared-Raman (TIRR) spectroscopy, drives THz frequency modes, and probes the effect of this excitation on O-H stretch modes through a probing step that upconverts the mid-IR into the visible. A heterodyne detection scheme allows phase-sensitive detection. This method is applied first to CaF₂, and then to water with varying isotopic composition.

The strength of this paper is the proposal of a new technique with novel new capabilities that will be of interest to those that study water as well as other molecular liquids and solids. This method may indeed be able to characterize the dynamics that they hope to study, however, this manuscript attempts to accomplish too much without enough characterization. The result is that the reader cannot judge the validity or usefulness of the results. The content, which spans describing and testing a new method, applying it to water dynamics, and describing molecular dynamics simulation methods, are really the content for three papers. Although I am very excited to learn more about this method, my reading leaves me unclear about the merits of the technique, and not convinced that the authors have been able to reveal something new. Certainly at this point it is difficult to follow, and does not properly justify its conclusions. It is my recommendation that the authors make the exposition of this method and explanation of its implications for water a more methodical and thoughtful exercise. Therefore I do not recommend publication.

The following questions and comments may help illustrate the type of issues that I struggled with:

One of the weaknesses of THz spectroscopy is that the THz waveforms are complex, and influence spectra in non-intuitive manners. Modeling and interpreting spectra typically require an understanding of this THz waveform (more than the spectrum). It is important to show the time-domain waveform (like Figure 2a) for comparison to water data, and to include it in calculations or simulations. I am left wondering how the positive and negative THz field oscillations influence the shape of the TIRR signal. Beyond that the measured time-domain response and absolute value frequency-domain spectra are not very intuitive.

Part of the difficulty is the multi-resonant nature of this technique, which is both the origin of its strength and complexity. It is hard to sort through singly and doubly resonant signals. The interpretation of isotope dilute water samples is also particularly difficult, since the THz spectrum will differ dramatically between these samples. Including linear THz and mid-IR absorption spectra of the isotope dilute samples is important for interpreting the data.

It is my recommendation that the authors use a different name for this new method, since the term "Raman" in the name confused me. "Raman" in my mind refers to driving low frequency excitations through difference frequency processes involving higher frequency fields, which I don't see here. It seems to bear more similarity to the DOVE methods of John Wright or three-photon sum frequency than a Raman technique.

Power dependences for the signal level on the input intensities would help convince that the authors are actually observing the signal they claim. Similarly a quantitative description of the intensity of the singly and doubly resonant signals would help describe the signal contributions.

Reviewer #2 (Remarks to the Author):

Grechko et al. carried out both experimental and MD simulation studies of water to elucidate vibrational couplings between high-frequency OH stretch modes and low-frequency intermolecular H-bond (stretch and bend) modes. As mentioned by the authors, it is very important to understand the detailed mechanism of vibrational energy relaxation of intramolecular high-

frequency modes of water in solution to understand the chemistry of water in general. However, a lack of experimental techniques that can be used to directly probe such couplings has been a serious limitation in quantitatively describing water vibrational relaxation. The authors have developed a novel technique that combines broadband THz, IR, and visible pulses to measure the vibration-vibration-electronic sum-frequency-generation field with an heterodyne-detection scheme. This reviewer believes that this technique is going to be a useful coherent multidimensional spectroscopic tool for studying various systems where the vibrational couplings between molecular vibrations and low-frequency THz-active modes play important roles in general.

The measured two-dimensional THz-IR-Raman spectra for a few different D2O/H2O mixed solutions in Figure 3 are highly interesting and informative. From the Fourier-transformed 2D spectra, one could easily identify the cross peaks that indicate direct mode-mode couplings between low-frequency H-bond (stretch and bend) modes and water O-H stretch modes. Although the MD simulation results do not match with the experimental spectra perfectly, the agreement can be considered to be at least qualitatively acceptable. This reviewer considers this work very interesting and recommend its publication in Nature Communications after minor revisions made.

(1) As discussed in Ref.35 and another paper (PHYSICAL REVIEW A, VOLUME 61, 023406, 2000), the vibrational couplings that can be measured with doubly (vibrationally) resonant IR-IR-vis (or THz-IR-vis) sum-frequency-generation (SFG) are induced by a few different mechanisms, i.e., transition dipole couplings, mechanical anharmonic couplings, etc. The authors concluded that (in page 10) the transition dipole couplings between low- and high-frequency modes are dominant, on the basis of their observation of the nodal line in the 2D spectra (Fig. 3k,l). Detailed discussion seems to be given in Supp Material, but it would be better to put it in the main text.

(2) Then, one major issue that this reviewer wants to raise is the connection between vibrational couplings and vibrational energy relaxation mechanism. The main goal of this work is to show that the vibrational energy relaxation between intramolecular high-frequency modes and intermolecular low-frequency modes can be caused by direct couplings. However, the fact that the two sets of modes are coupled via transition dipoles does not necessarily indicate that the vibrational energy of O-H stretch modes can be directly transferred to low-frequency H-bond modes. The THz-IR-vis SFG signal just indicates non-zero nonlinear transition probability, which is simply determined by the product of transition dipole of THz-mode, transition dipole of IR-active mode, and transition polarizability of the combination mode. It, non-zero nonlinear spectroscopic signal, actually does not mean that the vibrational energy relaxation would occur through such couplings effectively, since no experimental evidence on the strong anharmonic couplings between the two sets of modes has been observed here. The authors might want to clarify this point.

Reviewer #3 (Remarks to the Author):

In this paper, the authors reported the direct observation of the coupling between the intra- and intermolecular modes of liquid water using novel 2D THz-IR-Raman spectroscopy. Since this result should be related with the large body of the experimental and theoretical researchers of water, the impact of the present study would be high. Further extension of the present approach to the water in the different phase involving varieties of ice states are possible. In this sense, this manuscript has great potential to contribute to the field of chemical physics and physical chemistry.

My primary concern about the present paper is on the theoretical analysis of the experimentally obtained signal. Unlike 3DIR spectroscopy, the cross peak in the present spectroscopy are not necessary to be a mode-mode coupling peak. Thus, the theoretical analysis is significant to carry out this kind of measurements, while the theoretical descriptions in the present paper missing many of important aspects. I wish the authors to rebalance the theoretical descriptions with the experimental descriptions. After modifying this point, the paper should be published in nature communications.

Some more detailed comments:

-P2 introduction. 2D spectroscopy in this sort was first proposed by Ref. 40 in the authors article. Complexity of analyzing 2D signals was described in detail in that paper. This should be mentioned at the beginning of the paper. Otherwise, the readers may not be realized the difference between conventional 2D spectroscopies and the present 2D spectroscopy.

-P7. The second paragraph. The methodology of simulation should be explained in some more detail or at least its relevant references should be given. In particular, the authors' analysis seemed to be classical throughout the paper; this classical nature must be mentioned, because this might be the cause of the discrepancy between the experimental results and theoretical results. I am also wondering how the authors included 5% of D₂O in the simulation. Then, why the authors couldn't study the cases of higher D₂O concentrations. These points must be described.

-P10. The second paragraph. The origin of the cross peaks that attributed from nonlinear polarizability, nonlinear dipolar elements, and anharmonic coupling had been discussed in ref. 40 in detail. The analysis that the authors carried out in the supplementary work is poor and insufficient and poor, because the authors completely neglected the effects of vibrational dephasing for the OH mode, assumed unrealistic Morse potential for intermolecular modes, and neglected the anharmonic coupling between the inter- and intra-molecular modes. If the authors wish to include a simple argument, the approach by K. Okumura and Y. Tanimura, Chem. Phys. Lett. 278, 175 (1997) should be a better starting point. But, I suggest the authors to simply omit the supplementary discussions for this part, then just cite ref. 40.

-P12 It should be better for the authors to discuss a possibility to carry out 2D THz-Raman-IR measurements as well as 2D Raman THz-IR. By doing this kind of measurements, the authors can check the consistency of the measurements and analysis. These measurements may give new information for nonlinear polarizability and nonlinear dipole moments.

Reviewer #1:

This paper reports an interesting and potentially powerful new spectroscopy for studying the dynamics of water by revealing the coupling of high frequency OH stretch vibrations with low frequency modes of water.

We are grateful to the reviewer for the constructive remarks and appreciate the positive comment. We have addressed the reviewer's concerns in the revised manuscript, and provide a point-by-point reply in the following:

The method, termed terahertz-infrared-Raman (TIRR) spectroscopy, drives THz frequency modes, and probes the effect of this excitation on O-H stretch modes through a probing step that upconverts the mid-IR into the visible. A heterodyne detection scheme allows phase-sensitive detection. This method is applied first to CaF₂, and then to water with varying isotopic composition.

The strength of this paper is the proposal of a new technique with novel new capabilities that will be of interest to those that study water as well as other molecular liquids and solids. This method may indeed be able to characterize the dynamics that they hope to study, however, this manuscript attempts to accomplish too much without enough characterization. The result is that the reader cannot judge the validity or usefulness of the results. The content, which spans describing and testing a new method, applying it to water dynamics, and describing molecular dynamics simulation methods, are really the content for three papers. Although I am very excited to learn more about this method, my reading leaves me unclear about the merits of the technique, and not convinced that the authors have been able to reveal something new. Certainly at this point it is difficult to follow, and does not properly justify its conclusions. It is my recommendation that the authors make the exposition of this method and explanation of its implications for water a more methodical and thoughtful exercise. Therefore I do not recommend publication.

The following questions and comments may help illustrate the type of issues that I struggled with:

One of the weaknesses of THz spectroscopy is that the THz waveforms are complex, and influence spectra in non-intuitive manners. Modeling and interpreting spectra typically require an understanding of this THz waveform (more than the spectrum). It is important to show the time-domain waveform (like Figure 2a) for comparison to water data, and to include it in calculations or simulations. I am left wondering how the positive and negative THz field oscillations influence the shape of the TIRR

signal. Beyond that the measured time-domain response and absolute value frequency-domain spectra are not very intuitive.

We appreciate the reviewer's comment that THz spectroscopy, at the field level, is less straightforward than conventional spectroscopy at the intensity level. We would like to point out – in all modesty – that our group has extensive experience in dealing with and interpreting complex THz waveforms. Indeed, we've written the 2011 Reviews of Modern Physics paper on THz spectroscopy, which deals extensively with THz waveform analysis, and has been cited almost 500 times ("Carrier dynamics in semiconductors studied with time-resolved terahertz spectroscopy", R Ulbricht, E Hendry, J Shan, TF Heinz, M Bonn, Rev. Mod. Phys. 83 (2), 543, 2011). As such, we are well aware of potential artifacts in the measured spectra due to the variation of the time-domain profile of the THz pulse. To address the Reviewer's comment in the revised SI we provided the time-domain profile of the THz pulse and its intensity spectrum measured by the air-based coherent (ABC) detection (Supplementary Fig. 6c,d). We included the THz pulse in the calculations of the 2D TIRV spectra by convolution of the simulated nonlinear response function with the laser pulses (already in the original manuscript). We performed the convolution in the frequency domain which is given by the product of the spectra of the response function and the laser fields. In order to demonstrate this, we derived Supplementary Eq. (11) in the revised SI.

In principle, the 2D TIRV signals can indeed be considered as sum (interference) of the signals generated by the positive and negative oscillations of the THz field. We don't provide such consideration in this work because excitation of the LFM is a dynamic process in which a sample interacts with the entire THz pulse and the separation into positive and negative THz field periods doesn't provide a physically intuitive picture. We note that in this sense, THz pulses are no different from more "conventional" laser pulses at higher frequencies that also exhibit electromagnetic fields with both positive and negative components: it is the amplitude and frequency of the field oscillation that determine the material response to the field. We further note that we determine the time-domain waveform independently, and measure the data in the time domain, but a representation of that 2D TIRV data in the frequency domain (2D spectra) is more instructive because they are given by product of the third-order nonlinear response function $S^{(3)}$ of a sample and the spectra of the laser pulses (Supplementary have added Eq. (11) in the revised SI). In order to explain this we added the following text on page 7 of the revised manuscript:

“Because for water the decay time of the LFM coherence is comparable to the duration of the THz pulse the time-domain 2D TIRV data are less intuitive than the frequency-domain 2D TIRV spectra. A 2D TIRV spectrum represents the product of the spectrum of the third-order nonlinear response function and the spectra of the laser pulses (Supplementary Eq. (11)).”

Part of the difficulty is the multi-resonant nature of this technique, which is both the origin of its strength and complexity. It is hard to sort through singly and doubly resonant signals. The interpretation of isotope dilute water samples is also particularly difficult, since the THz spectrum will differ dramatically between these samples. Including linear THz and mid-IR absorption spectra of the isotope dilute samples is important for interpreting the data.

Indeed, as with many non-linear spectroscopies, non-resonant (or, in this case, singly resonant) signals complicate the measurements. In principle, the resonant and non-resonant responses can be disentangled by resolving the phase of the signal. Such an approach requires careful phasing of the 2D TIRV signals which is the subject for the future work. Here we experimentally distinguish between the singly- and doubly-resonant signals by comparing the response of different O-H concentrations with that of D₂O. For pure D₂O, the IR at O-H stretch frequencies is not resonant. The resonance with the low-frequency mode remains, given the invariance of the LFM to H/D substitution, as reflected by the far-IR absorption spectra in Fig. 1a.

We agree that both the THz (far-infrared) and mid-IR absorption spectra are important, and have shown these in Fig. 1a,b (already in the original manuscript). For the two boundary cases, 100% H₂O and 100% D₂O, far-IR absorption spectra evidence only very minor changes of the absorbance (and thus the LFM) in the THz frequency range of this work (20-450 cm⁻¹) upon a change of the isotopomer. These results are in good agreement with other publications (for example The Journal of Chemical Physics 131, 184505 (2009), see reproduced Figure below). The change in the far-IR absorption spectrum is characterized mainly by the red-shift of the librational mode (centered at ~670 cm⁻¹ for H₂O and ~490 cm⁻¹ for D₂O) which results in the ~15% increase of the absorbance at the blue side of the THz spectral range of our experiment. Absorption spectra for other H/D ratios must necessarily vary between these two, very similar, limiting cases.

J. Chem. Phys. **131**, 184505 (2009)

Reprinted from The Journal of Chemical Physics 131, 184505 (2009),
with the permission of AIP Publishing

The invariance of the LFM to the isotope dilution is used in our analysis and is reflected by the following statement on page 8 of the (original) manuscript:

“Because the change of the LFM with isotope dilution is minute (Fig. 1a), the drastic change of the $\langle S_{OH}^{(3)} \rangle$ must be attributed to a change in the character of the O-H stretch vibrations.”

The mid-IR absorption spectra for 5% H/D, 20% H/D, 50% H/D and 100% H₂O are shown in the Fig. 1b and the changes in these spectra upon isotope dilution are discussed on page 8 of the (original) manuscript:

“Increasing concentration of the O-H groups begets two new types of vibrational states. Firstly, within each H₂O molecule, the O-H stretch is mixed with the overtone vibration of the H-O-H bending mode because of the Fermi resonance (FR). This resonance is appreciably weaker for HOD because of the large energy mismatch of the two levels⁴². Thus, increasing number of H₂O species enhances signal generated by the FR. Secondly, the increasing density of the O-H groups induces the formation of the

delocalized O-H stretch exciton because of intermolecular coupling. As a result, the change of the $\langle S_{OH}^{(3)} \rangle$ can be linked to the appearance of the FR and vibrational exciton formation.”

It is my recommendation that the authors use a different name for this new method, since the term "Raman" in the name confused me. "Raman" in my mind refers to driving low frequency excitations through difference frequency processes involving higher frequency fields, which I don't see here. It seems to bear more similarity to the DOVE methods of John Wright or three-photon sum frequency than a Raman technique.

We agree with the Reviewer that the use of the term “Raman” can potentially confuse readers and we follow the Reviewer’s recommendation. We change the name of the spectroscopy to two-dimensional terahertz-infrared-visible (2D TIRV) throughout the manuscript and Supplementary Information. In general, the spectroscopy developed in this work can indeed be considered as an extension of the 2D DOVE spectroscopy of John Wright to the THz-IR frequency range, and in order to explain this we have included the following statement on page 3 of the manuscript:

“This type of spectroscopy was recently proposed by Ito and Tanimura³³ and can be considered as an extension of the 2D-DOVE (IIV-SFG)^{18,34} spectroscopy to the THz-IR spectral range.”

Power dependences for the signal level on the input intensities would help convince that the authors are actually observing the signal they claim. Similarly a quantitative description of the intensity of the singly and doubly resonant signals would help describe the signal contributions.

In our experiment, the change of the intensity of the 2D TIRV signal for different intensities of the IR and VIS pulses cannot be disentangled from the change of the intensity of the local oscillator because the local oscillator is generated by SFG mixing of the IR and VIS pulses. This hinders intensity dependence measurements for the IR and VIS pulses. However, in our measurements, we detect the 2D TIRV signal at the sum frequency of the IR and VIS pulses ($\omega_{SIG} = \omega_{IR} + \omega_{VIS}$). As such, these fields necessarily can only interact once each with the sample; otherwise the signal would appear at $\omega_{SIG} = n \cdot \omega_{IR} + m \cdot \omega_{VIS}$, with n, m different from 1. Following the reviewer’s suggestion, we have measured dependence of the intensity of the 2D TIRV signal on the intensity of the THz pulse. The new data shown in Supplementary Fig. 2 demonstrates the linear dependence of the signal on the THz field strength, which provides further evidence for the four-wave mixing nature of the water

signal. To describe the dependence of the 2D TIRR signal on the IR, VIS and THz fields we have added the following text on page 7 of the revised manuscript:

“For the water signals depicted in Fig. 3a-e to be generated by FWM of THz, IR and VIS pulses it must be produced by one interaction of the sample with each of these fields. Because we detect the signal at the sum frequency of the IR and VIS laser pulses ($\omega_{VIS} + \omega_{IR} \pm \omega_{THz}$, $\sim 15800 \text{ cm}^{-1}$) the signal must indeed be generated by one interaction with each of the IR and VIS fields. The linear dependence of the signal on the strength of the THz field (Supplementary Fig. 2) confirms that a sample interacts only once with the THz pulse, in line with these 2D TIRV signals being generated by FWM of the THz, IR and VIS fields in water.”

We agree that quantitative description of intensity variation of the singly- and doubly-resonant signals is necessary to describe their contributions to the total 2D TIRV signal for different H/D mixtures. We consider this variation in the Supplementary Discussion section “Change of the 2D TIRV signal intensity with sample composition” (already in the original SI) and use it to derive the 2D TIRV spectrum for localized O-H stretch oscillators in Fig. 4a (already in the original manuscript).

Reviewer #2:

Grechko et al. carried out both experimental and MD simulation studies of water to elucidate vibrational couplings between high-frequency OH stretch modes and low-frequency intermolecular H-bond (stretch and bend) modes. As mentioned by the authors, it is very important to understand the detailed mechanism of vibrational energy relaxation of intramolecular high-frequency modes of water in solution to understand the chemistry of water in general. However, a lack of experimental techniques that can be used to directly probe such couplings has been a serious limitation in quantitatively describing water vibrational relaxation. The authors have developed a novel technique that combines broadband THz, IR, and visible pulses to measure the vibration-vibration-electronic sum-frequency-generation field with an heterodyne-detection scheme. This reviewer believes that this technique is going to be a useful coherent multidimensional spectroscopic tool for studying various systems where the vibrational couplings between molecular vibrations and low-frequency THz-active modes play important roles in general.

The measured two-dimensional THz-IR-Raman spectra for a few different D2O/H2O mixed solutions in Figure 3 are highly interesting and informative. From the Fourier-transformed 2D spectra, one could easily identify the cross peaks that indicate direct mode-mode couplings between low-frequency H-bond (stretch and bend) modes and water O-H stretch modes. Although the MD simulation results do not match with the experimental spectra perfectly, the agreement can be considered to be at least qualitatively acceptable. This reviewer considers this work very interesting and recommend its publication in Nature Communications after minor revisions made.

We are grateful to the reviewer for the constructive remarks and appreciate the positive comment. We have addressed the reviewer's concerns in the revised manuscript, and provide a point-by-point reply in the following:

(1) As discussed in Ref.35 and another paper (PHYSICAL REVIEW A, VOLUME 61, 023406, 2000), the vibrational couplings that can be measured with doubly (vibrationally) resonant IR-IR-vis (or THz-IR-vis) sum-frequency-generation (SFG) are induced by a few different mechanisms, i.e., transition dipole couplings, mechanical anharmonic couplings, etc. The authors concluded that (in page 10) the transition dipole couplings between low- and high-frequency modes are dominant, on the basis of their observation of the nodal line in the 2D spectra (Fig. 3k,l). Detailed discussion seems to be given in Supp Material, but it would be better to put it in the main text.

We appreciate the Reviewer's suggestion, and had indeed included a discussion along those lines in the SI of the original submission. However, Reviewer #3 in his/her comment P10, questions the usefulness of such a simple model, and the conclusion obtained therewith. We follow the suggestion of the Reviewer #3 and have accordingly removed this model and the corresponding discussion from the manuscript and Supplementary

Information. Instead, we refer the work by Ito and Tanimura (Ref. 33 in the revised manuscript) who have carried out a detailed analysis of the nature of the coupling between the LFM and O-H stretch using a similar model for the MD calculations. This has resulted in the following changes to the text:

We have deleted the following paragraph on page 10 of the manuscript:

“In principle, three distinct feasible mechanisms can give rise to the coupling: polarizability coupling, dipole moment coupling and mechanical potential coupling^{39,40}. We can distinguish between these three potential contributions by examining the analytical expression of the three-body response function responsible for the signal. This analysis, detailed in the Supplementary Discussion, reveals that the three contributions have distinctly different 2D profiles (Supplementary Fig. 3). The presence of the nodal line in the 2D spectra (Fig. 3k,l) reveals that the main contribution arises from the coupling between the transition dipole moments of the low frequency and high frequency modes. The ω_2 frequency of the nodal line in the 2D TIRR spectrum corresponds to the maximum of the nonlinear optical susceptibility of the sample.”

We have deleted the section “Analytical expressions for the 2DTIRR response function of two classical coupled Morse Oscillators” from the Supplementary Discussion (pages 6-7).

We have added the following statement on page 12 of the revised manuscript:

“Our findings are in good agreement with the recent theoretical predictions by Ito and Tanimura using the same MD model³³. Their analysis shows that for water both electrical and mechanical anharmonicities contribute significantly to the coupling.”

(2) Then, one major issue that this reviewer wants to raise is the connection between vibrational couplings and vibrational energy relaxation mechanism. The main goal of this work is to show that the vibrational energy relaxation between intramolecular high-frequency modes and intermolecular low-frequency modes can be caused by direct couplings. However, the fact that the two sets of modes are coupled via transition dipoles does not necessarily indicate that the vibrational energy of O-H stretch modes can be directly transferred to low-frequency H-bond modes. The THz-IR-vis SFG signal just indicates non-zero nonlinear transition probability, which is simply determined by the product of transition dipole of THz-mode, transition dipole of IR-active mode, and transition polarizability of the combination mode. It, non-zero nonlinear spectroscopic signal, actually does not mean that the vibrational energy relaxation would occur through such couplings effectively, since no experimental

evidence on the strong anharmonic couplings between the two sets of modes has been observed here. The authors might want to clarify this point.

We agree with the reviewer that the coupling we observe could result from only the presence of electrical (and not mechanical) anharmonicity between the LFM and O-H stretch vibration, and as such doesn't necessarily result in a pathway for efficient energy relaxation. As described above, and included in the revised manuscript, the detailed analysis by Ito and Tanimura (Ref. 33 in the revised manuscript) using similar water model for the MD calculations demonstrated that the coupling is caused by both the mechanical and electrical anharmonicities. Therefore, the direct coupling between the O-H stretch and LFM vibrations measured in this work provides a competing channel for the vibrational energy relaxation. We have changed the text on page 12 accordingly:

“Strong **mechanical** coupling between the O-H stretch and the LFM manifests that in the liquid the low-frequency intermolecular and high-frequency intramolecular motions generate cooperative motion of the atoms. Such cooperative motion provides a competing channel for direct non-adiabatic energy dissipation to collective, delocalized intermolecular modes.”

Reviewer #3:

In this paper, the authors reported the direct observation of the coupling between the intra- and intermolecular modes of liquid water using novel 2D THz-IR-Raman spectroscopy. Since this result should be related with the large body of the experimental and theoretical researchers of water, the impact of the present study would be high. Further extension of the present approach to the water in the different phase involving varieties of ice states are possible. In this sense, this manuscript has great potential to contribute to the field of chemical physics and physical chemistry.

We are grateful to the reviewer for the constructive remarks and appreciate the positive comment. We have addressed the reviewer's concerns in the revised manuscript, and provide a point-by-point reply in the following:

My primary concern about the present paper is on the theoretical analysis of the experimentally obtained signal. Unlike 3DIR spectroscopy, the cross peak in the present spectroscopy are not necessary to be a mode-mode coupling peak. Thus, the theoretical analysis is significant to carry out this kind of measurements, while the theoretical descriptions in the present paper missing many of important aspects. I wish the authors to rebalance the theoretical descriptions with the experimental descriptions. After modifying this point, the paper should be published in nature communications.

Some more detailed comments:

-P2 introduction. 2D spectroscopy in this sort was first proposed by Ref. 40 in the authors article. Complexity of analyzing 2D signals was described in detail in that paper. This should be mentioned at the begging of the paper. Otherwise, the readers may not be realized the difference between conventional 2D spectroscopies and the present 2D spectroscopy.

We appreciate this comment because it stimulated us to better explain the origin of the off-diagonal peaks in the 2D TIRV spectroscopy (we have changed the name of the 2D spectroscopy presented here to 2D TIRV). As discussed by Ito and Tanimura in Ref. 33 of our revised manuscript the off-diagonal peaks in the 2D IIR spectroscopy can be generated by excitation of the overtone and population states of the same vibrational mode. Obviously, such off-diagonal peaks do not represent coupling between different vibrations. But in the 2D TIRV spectroscopy excitation of the overtone and population vibrational states is impossible because of the very different frequencies of the THz and IR laser pulses. In order to explain this, we have included the following text on page 4 of the revised manuscript:

"We note that in general the off-diagonal peaks in a doubly vibrationally enhanced spectroscopy can also be generated by excitation of the

overtone and population states of the same mode^{33,39}, but in 2D TIRV spectroscopy such excitation pathways are impossible due to the large frequency difference between the THz and IR fields. Thus, the off-diagonal peaks in the 2D TIRV spectra reflect coupling between different modes.”

-P7. The second paragraph. The methodology of simulation should be explained in some more detail or at least its relevant references should be given. In particular, the authors' analysis seemed to be classical throughout the paper; this classical nature must be mentioned, because this might be the cause of the discrepancy between the experimental results and theoretical results. I am also wondering how the authors included 5% of D₂O in the simulation. Then, why the authors couldn't study the cases of higher D₂O concentrations. These points must be described.

In the original manuscript, the methodology of the calculations was given in the Methods section. To address the Reviewer's comment and make it easier to understand for the readers we have moved this description into the Results section. In the revised manuscript the text on page 7 reads:

“Figures 3k,l show the absolute-value 2D TIRV spectra obtained by classical molecular dynamics (MD) calculations for the 5% H/D and 100% H₂O samples, respectively. The calculations of the nonlinear optical susceptibility are performed analogously to those in Ref. ³³. In brief, we use POLI2VS water model⁴¹ for 64 H₂O and 58 D₂O + 6 HOD molecules representing the 100% H₂O and 5% H/D samples, respectively. After 100 ps equilibration MD runs under the constant temperature conditions at 300 K, the time domain 2D TIRV response functions are calculated for the time periods of 250 fs using the non-equilibrium-equilibrium hybrid response function algorithm using a total of 10⁶ non-equilibrium trajectories. The 2D TIRV spectra are generated by Fourier transforming the time-domain response functions and subsequent convolution in the frequency domain with the THz and IR pulses of the experiment to allow direct comparison with the experimental results (see Supplementary Methods for more details on calculations and Supplementary Fig. 8 for the spectra without the convolution). Because the calculations are time-demanding, we report here the spectra for only two samples which represent the limiting cases of very low (5%) and very high (100%) O-H concentration.”

-P10. The second paragraph. The origin of the cross peaks that attributed from nonlinear polarizability, nonlinear dipolar elements, and anharmonic coupling had been discussed in ref. 40 in detail. The analysis that the authors carried out in the supplementary work is poor and insufficient and poor, because the authors completely neglected the effects of vibrational dephasing for the OH mode, assumed

unrealistic Morse potential for intermolecular modes, and neglected the anharmonic coupling between the inter- and intra-molecular modes. If the authors wish to include a simple argument, the approach by K. Okumura and Y. Tanimura, Chem. Phys. Lett. 278, 175 (1997) should be a better starting point. But, I suggest the authors to simply omit the supplementary discussions for this part, then just cite ref. 40.

We appreciate this comment because it points to a critical oversimplification in our model. We follow the Reviewer's suggestion and dismiss the model and its discussion. This has resulted in the following changes to the text:

We have deleted the following paragraph on page 10 of the manuscript:

“In principle, three distinct feasible mechanisms can give rise to the coupling: polarizability coupling, dipole moment coupling and mechanical potential coupling^{39,40}. We can distinguish between these three potential contributions by examining the analytical expression of the three-body response function responsible for the signal. This analysis, detailed in the Supplementary Discussion, reveals that the three contributions have distinctly different 2D profiles (Supplementary Fig. 3). The presence of the nodal line in the 2D spectra (Fig. 3k,l) reveals that the main contribution arises from the coupling between the transition dipole moments of the low frequency and high frequency modes. The ω_2 frequency of the nodal line in the 2D TIRR spectrum corresponds to the maximum of the nonlinear optical susceptibility of the sample.”

We have deleted the section “Analytical expressions for the 2DTIRR response function of two classical coupled Morse Oscillators” from the Supplementary Discussion (pages 6-7).

We have added the following statement on page 12 of the revised manuscript:

“Our findings are in good agreement with the recent theoretical predictions by Ito and Tanimura using the same MD model³³. Their analysis shows that for water both electrical and mechanical anharmonicities contribute significantly to the coupling.”

-P12 It should be better for the authors to discuss a possibility to carry out 2D THz-Raman-IR measurements as well as 2D Raman THz-IR. By doing this kind of measurements, the authors can check the consistency of the measurements and analysis. These measurements may give new information for nonlinear polarizability and nonlinear dipole moments.

The reviewer raises a very interesting point, and we fully agree with the Reviewer that the complementary 2D THz-Raman-IR and 2D Raman-THz-

IR measurements would be beneficial to further elucidate the coupling between the O-H stretch and the LFM in water. Unfortunately, however, the implementation of such spectroscopy techniques requires a significantly different experimental approach from the one presented in this work, and really is outside the scope of the present work. Their feasibility is yet to be tested.

Reviewer #2 (Remarks to the Author):

The authors have addressed all the issues raised by this reviewer and revised the manuscript and SI accordingly. I would like to recommend its publication in Nat. Comm. as is.

Reviewer #3 (Remarks to the Author):

This paper reports a new significant spectroscopy technique for studying the dynamics of molecular liquids including water. In the revised manuscript, the authors' properly responded all of the questions raised by the reviewers and the readability of the paper improved significantly. Therefore, I recommend the present paper to be published as is.

Reviewer #4 (Remarks to the Author):

The major point of referee 1 has been:

"The strength of this paper is the proposal of a new technique with novel new capabilities that will be of interest to those that study water as well as other molecular liquids and solids. This method may indeed be able to characterize the dynamics that they hope to study, however, this manuscript attempts to accomplish too much without enough characterization. The result is that the reader cannot judge the validity or usefulness of the results. The content, which spans describing and testing a new method, applying it to water dynamics, and describing molecular dynamics simulation methods, are really the content for three papers. Although I am very excited to learn more about this method, my reading leaves me unclear about the merits of the technique, and not convinced that the authors have been able to reveal something new. Certainly at this point it is difficult to follow, and does not properly justify its conclusions. It is my recommendation that the authors make the exposition of this method and explanation of its implications for water a more methodical and thoughtful exercise. Therefore I do not recommend publication."
and with the exception of the last sentence, I agree 100%. This is very tough stuff, which is in the nature of things, and it is currently not clear, I assume also not to the authors, what exactly those 2D responses can tell us. But that should not be seen as a negative, rather the authors should be congratulated for trying out something truly new. Only future will show whether this new technique pays off. I certainly can see the potential in that technique, just like the referee.

Referee 1 continues to illustrate his/her criticism based on a few concrete questions. The authors try their best to answer them, however that does not really solve the problem "that the reader cannot judge the validity or usefulness of the results". But, again, I think that this is the nature of things and should not be taken as a negative.

Overall, I highly recommend publication of this paper.